# DYNAMICS-AWARE EMBEDDINGS

**William F. Whitney**[1], **Rajat Agarwal**[1], **Kyunghyun Cho**[13], **and Abhinav Gupta**[23]

[1]Department of Computer Science, New York University
[2]Robotics Institute, Carnegie Mellon University
[3]Facebook AI Research
wwhitney@cs.nyu.edu

## ABSTRACT

In this paper we consider self-supervised representation learning to improve sample efficiency in reinforcement learning (RL). We propose a forward prediction objective for simultaneously learning embeddings of states and action sequences. These embeddings capture the structure of the environment's dynamics, enabling efficient policy learning. We demonstrate that our action embeddings alone improve the sample efficiency and peak performance of model-free RL on control from low-dimensional states. By combining state and action embeddings, we achieve efficient learning of high-quality policies on goal-conditioned continuous control from pixel observations in only 1-2 million environment steps.

## 1  INTRODUCTION

In recent years, there has been a lot of excitement around end-to-end model-free reinforcement learning for control, both in simulation (Lillicrap et al., 2015; Andrychowicz et al., 2018; Haarnoja et al., 2018b; Fujimoto et al., 2018) and on real hardware (Kalashnikov et al., 2018; Haarnoja et al., 2018c). In this paradigm, we simultaneously learn intermediate representations and policies by maximizing rewards provided by environment. End-to-end learning has one indisputable advantage: since every component of the system is optimized for the end objective, there are no sub-optimal modules that limit best-case performance by losing task-relevant information.

Learning only from the target task is however a double-edged sword. When the end objective provides only weak signal for learning, a policy with a poor representation may require many samples to learn a better one. By contrast, a policy with a good representation may be able to rapidly fit a simple function of that representation even with weak signal.

Consider the environment shown in Figure 1, and two representations of its state: coordinates and pixels. As a function of the agent's $x$ coordinate, the value function is simple and smooth. The coordinate representation has structure which is useful for learning about the task; namely, points which are close in $L^2$ distance have similar values. By contrast, a pixel representation of the agent's state (below, blue) is practically a one-hot vector. Two states whose $x$ coordinates differ by one unit have pixels exactly as different as states which differ by 100 units. This illustrates the importance of good representations and the potential of representation learning to aid RL.

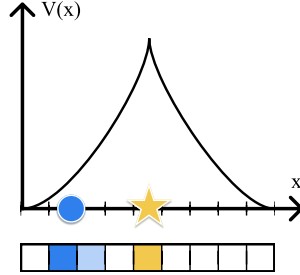

Figure 1: A 1D environment. The agent (blue dot) can move continuously left and right to reach the goal (gold star).

We propose a self-supervised objective for learning embeddings of states and action sequences such that a pair of states or action sequences will be close together if they have similar outcomes. This objective simultaneously trains a smooth embedding space for states and a temporally abstract action space for control which is task-independent and generalizes across goals and objects.

We demonstrate the effectiveness of our representation learning objective by training the twin delayed deep deterministic policy gradient algorithm (TD3) (Fujimoto et al., 2018) with learned action and state spaces. With a learned representation of temporally abstract actions, our method

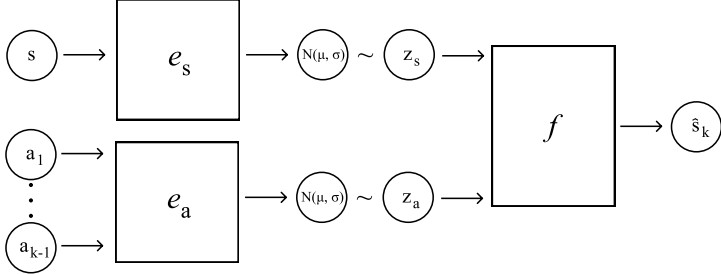

Figure 2: Computational architecture for training the DynE encoders $e_a$ and $e_s$. The encoders are trained to minimize the information content of the learned embeddings while still allowing the predictor $f$ to make accurate predictions.

exhibits improved sample efficiency compared to state-of-the-art RL methods on control tasks, with larger gains on more complex environments. When additionally combined with our learned state representation, our method allows TD3 to scale to pixel observations. We demonstrate good performance on a simple family of goal-conditioned 2D control tasks within a few million environment steps without adjusting any TD3 hyperparameters. This stands in contrast to end-to-end model-free RL from pixels, which requires extensive tuning (Lillicrap et al., 2015) and on the order of 100 million environment steps[1] (Barth-Maron et al., 2018).

## 2 DYNAMICS-AWARE EMBEDDINGS

### 2.1 NOTATION

We consider the framework of reinforcement learning in Markov decision processes (MDPs).[2] We denote the state of an environment (e.g. joint angles of a robot or pixels) by $s \in \mathcal{S}$, and we assume that the states given by the environment satisfy the Markov property. We refer to a sequence of actions $\{a_1, ..., a_k\} \in \mathcal{A}^k$ using the shorthand $\boldsymbol{a}^k$. We use $s' \sim \mathbf{T}(s, a)$ to refer to the environment's (stochastic) transition function, and overload it to accept sequences of actions: $s_{t+k} \sim \mathbf{T}(s_t, \boldsymbol{a}_t^k)$.

### 2.2 MODEL AND LEARNING OBJECTIVE

We propose that a good representation for reinforcement learning should represent states or actions close together if they have similar outcomes (resulting trajectories). This allows the agent to generalize from a small number of samples since each sample accurately reflects the value of all the states or actions in its neighborhood. In a Markov decision process the outcome of taking an action $a$ in a state $s$ is summarized by the distribution of resulting states $p(s'|s, a) = \mathbf{T}(s, a)$. Therefore we construct a method which embeds states and actions such that nearby embeddings have similar distributions of next states.

Our method, which we call Dynamics-aware Embedding (DynE), learns encoders $e_s$ and $e_a$ which embed a state and action sequence into latent spaces $z_s \in \mathcal{Z}_s$ and $z_a \in \mathcal{Z}_a$ respectively. These encodings are optimized to form a maximally compressed representation of the sufficient statistics of $p(s'|s, \boldsymbol{a}^k)$ such that $p(s'|s, \boldsymbol{a}^k) \approx p(s'|z_s, z_a)$. We approximate this by maximizing the following objective:

$$\mathcal{L}(\phi_s, \phi_a, \theta) = \underset{s, \boldsymbol{a}^k, s' \sim \rho^\pi}{\mathbb{E}} \Big[ -\log p(s'|z_s, z_a; \theta) \qquad \text{predict } s' \qquad (1)$$
$$+ \beta D_{\mathrm{KL}}\big(e_s(s; \phi_s) \,||\, \mathcal{N}(0, \boldsymbol{I})\big) \qquad \text{compress } s \qquad (2)$$
$$+ \gamma D_{\mathrm{KL}}\big(e_a(\boldsymbol{a}^k; \phi_a) \,||\, \mathcal{N}(0, \boldsymbol{I})\big)\Big] \qquad \text{compress } \boldsymbol{a}^k \qquad (3)$$

---

[1] Number of steps required to train D4PG taken from Hafner et al. (2018), as Barth-Maron et al. (2018) does not include this information.

[2] In the interest of space we omit the usual recap of Markov decision processes and reinforcement learning. We refer the reader to Section 2 of Silver et al. (2014) for notation and background on MDPs.

where $z_s \sim e_s(s)$, $z_a \sim e_a(\boldsymbol{a}^k)$, and $\rho^\pi$ is the distribution of transitions under a behavior policy $\pi$.

The DynE objective is similar to a $\beta$-VAE (Higgins et al., 2017a) for $s'$ but with a different variational family; like a $\beta$-VAE, it forms a variational lower bound on $p(s')$ when $\beta = \gamma = 1$. Where a variational autoencoder (Kingma & Welling, 2013; Rezende et al., 2014) or $\beta$-VAE chooses the variational family to be $\mathcal{Q} = \{q(z|s')\}$, we use a factored latent space $\{z_s, z_a\}$ and independent posterior approximations given the previous state and the action: $\mathcal{Q} = \{(q(z_s|s), q(z_a|\boldsymbol{a}^k))\}$. This factorization yields separate encoders for states and actions where the state encoder's output is valid for any action and vice versa.

The DynE objective can also be interpreted in the information bottleneck (IB) framework (Tishby et al., 2000). In the IB framework term (1) is the prediction objective and terms (2) and (3) regularize the latent representation to remove all extraneous information. Our construction is nearly identical to the approximate information bottleneck proposed by Alemi et al. (2016), with the main difference being the factorization of the representation into separate state and action components.

In our experiments we use an isotropic Normal distribution for $p(s'|z_s, z_a; \theta)$ such that term (1) reduces to $\|f(z_s, z_a; \theta) - s'\|_2^2$ where $f$ computes the mean. We use diagonal-covariance Normal distributions for $e_s$ and $e_a$ such that $\{\mu_s, \sigma_s^2\} = e_s(s)$, $\{\mu_a, \sigma_a^2\} = e_a(\boldsymbol{a}^k)$, $z_s \sim \mathcal{N}(\mu_s, \sigma_s^2)$, and $z_a \sim \mathcal{N}(\mu_a, \sigma_a^2)$. The behavior policy we use for data collection is $\pi = Unif(\mathcal{A})$.

## 3 USING LEARNED EMBEDDINGS FOR REINFORCEMENT LEARNING

### 3.1 DECODING TO RAW ACTIONS

In order to be useful for RL, the abstract action space produced by the encoder must be decodeable to raw actions in the environment. Since the mapping from action sequences to high-level actions is many-to-one, inverting it is nontrivial. We simplify this ill-posed problem by defining an objective with a single optimum.

Once the action encoder $e_a$ is fully trained, we hold it fixed and train an action decoder $d_a$ to minimize

$$\mathcal{L}(d_a) = \mathop{\mathbb{E}}_{z_a \sim \mathcal{N}(0, \boldsymbol{I})} \left[ \|e_a(d_a(z_a)) - z_a\|_2^2 + \lambda \|d_a(z_a)\|_2^2 \right] \tag{4}$$

The first term of this objective ensures that the action decoder $d$ is a one-sided inverse of $e_a$; that is, $e_a(d_a(z_a)) = z_a$ but $d_a(e_a(a_1, ..., a_k)) \neq a_1, ..., a_k$. The second term of the loss ensures that $d_a$ is in particular the minimum-norm one-sided inverse of $e_a$ and gives the objective for the output of $d_a$ a single minimum. Out of all the action sequences which have the same outcome, the minimum-norm sequence is desireable as it leads to trajectories which are smooth and consume less energy. We choose $\lambda$ to be small (e.g. $10^{-2}$) to ensure that the reconstruction criterion dominates the optimization.

### 3.2 EFFICIENT RL WITH TEMPORAL ABSTRACTION

Once equipped with a decoder which maps from high-level actions to sequences of raw actions, we train a high-level policy that solves a task by selecting high-level actions. In this section we extend the deterministic policy gradient (Silver et al., 2014) family of algorithms to work with temporally-extended actions while maintaining off-policy updates and learning from every environment step. This allows our method to achieve superior sample efficiency when working with high-level actions. In particular, we extend the twin delayed deep deterministic policy gradient (TD3) algorithm (Fujimoto et al., 2018) to work with the DynE representation of actions to form an algorithm we call DynE-TD3.

We first describe why DPG requires modifications to accommodate temporally-abstracted actions. One simple approach to combining DynE with DPG would be to incorporate the $k$-step DynE action space into the environment to form a new MDP. This MDP allows the use of DPG without modification; however, it only emits observations once every $k$ timesteps. As a result, after $N$ steps in the original environment, the deterministic policy $\mu$ and critic function $Q$ can only be trained on $N/k$ observations. This has a substantial impact on sample efficiency when measured in the original environment.

Instead we require an algorithm which can perform updates to the policy $\mu$ and critic $Q$ for every environment step. To do this, we train both $\mu$ and $Q$ in the abstract action space with minor changes to their updates. We distinguish these functions which use DynE actions from their raw equivalents by adding a superscript DynE, i.e. $\mu^{\text{DynE}}$ and $Q^{\text{DynE}}$. We augment the critic function with an additional input, $i$, which represents the number of steps $0 \leq i < k$ of the current embedded action $z$ that have already been executed. This forms the DynE-TD3 critic:

$$Q^{\text{DynE}}(e_s(s_t), z_t, i) = \sum_{j=0}^{k-i-1} \left(\gamma^j r_{t+j}\right) + \gamma^{k-i} Q^{\text{DynE}}\Big(e_s(s_{t+k-i}), \mu^{\text{DynE}}(e_s(s_{t+k-i})), 0\Big) \quad (5)$$

In plain language, the value of being on step $i$ of abstract action $e_t$ is the value of finishing the remaining $(k-i)$ steps of $z_t$ and then continuing on following the policy. This is similar to the idea of $k$-step returns (Sutton & Barto, 2018), but with a variable $k$ which depends on the step within the current plan. Whereas $k$-step returns would typically require an off-policy correction such as Retrace (Munos et al., 2016), conditioning on $z_t$ and $i$ determines all $k-i$ actions in the return. In effect, they remain a single action, making the update valid off policy. The DynE critic is trained by minimizing the Bellman error implied by eq. (5).

To update the policy we follow the standard DPG technique of using the gradient of the critic. We modify the algorithm to take into account that $i = 0$ at the time of issuing a new high-level action. The gradient of the return with respect to the policy parameters is then

$$\nabla_\theta J_\pi(\mu_\theta^{\text{DynE}}) \approx \mathop{\mathbb{E}}_{s \sim \rho^\pi} \left[\nabla_\theta \mu_\theta^{\text{DynE}}(e_s(s)) \, \nabla_z Q^{\text{DynE}}(e_s(s), z, 0)\big|_{z=\mu_\theta^{\text{DynE}}(e_s(s))}\right] \quad (6)$$

given that data was collected according to a behavior policy $\pi$.

## 4 RELATED WORK

Successor representations, an inspiration for this work, represent a state by the expected rate of future visits to other states (Dayan, 1993; Kulkarni et al., 2016b; Barreto et al., 2017). Successor representations have been demonstrated to be an effective model of animal and human learning (Momennejad et al., 2017; Stachenfeld et al., 2017). They are also one of the earliest realizations of the idea of representing each state by its future. Whereas successor representations learn future occupancy maps for a particular policy, we learn an embedding space where states are close together if they have similar outcomes for any policy.

Several papers have proposed using (variational) auto-encoders to learn embeddings for observations (Lange & Riedmiller, 2010; Van Hoof et al., 2016; Higgins et al., 2017b; Caselles-Dupré et al., 2018); unlike our work, these models operate on a single observation at a time and do not depend on the environment dynamics. Forward prediction has also been used as an auxiliary task to speed RL training (Jaderberg et al., 2016), and Jonschkowski et al. (2017) learn representations which adhere to physical constraints. Ghosh et al. (2018) propose to learn state embeddings using the action distribution of a goal-conditioned policy; however, their technique depends on already having a successful policy. Other work has proposed to use mutual information maximization to learn embeddings which facilitate exploration via intrinsic motivation (Kim et al., 2018).

Similarly to this work, hierarchical reinforcement learning seeks to learn temporal abstractions. These abstractions are variously defined as skills (Florensa et al., 2017; Hausman et al., 2018), options (Sutton et al., 1999; Bacon et al., 2017), or goal-directed sub-policies (Kulkarni et al., 2016a; Vezhnevets et al., 2017). Most closely related are SeCTAR (Co-Reyes et al., 2018) and HIRO (Nachum et al., 2018). SeCTAR simultaneously learns a generative model of future states and a low-level policy which can reach those states. HIRO learns a representation of goals such that a high-level policy can induce any action in a low-level policy. Unlike this work, both SeCTAR and HIRO learn state-dependent low-level policies, not action representations. Furthermore SeCTAR assumes the reward function is given ahead of time, and HIRO's off-policy performance depends on an approximate re-labeling of action sequences to train the high-level policy.

Also related are methods which attempt to learn embeddings of single actions to enable efficient learning in very large action spaces (Dulac-Arnold et al., 2015; Chandak et al., 2019). In particular, Chandak et al. (2019) learns a latent space of actions based on the effects of an action on the environment. However, their latent spaces are for a single action and they do not consider learned state representations. Another related direction is learning embeddings of one or more actions from demonstrations (Tennenholtz & Mannor, 2019); this embedded action space builds in prior knowledge from the demonstrator and can allow faster learning.

## 5 REPRESENTATION EXPERIMENTS

In this section we empirically investigate how the learned DynE representations reshape the problem of reinforcement learning. First we make a connection between temporal abstraction and exploration, revealing that DynE actions result in better state coverage. Then we probe the relationship between DynE state embeddings and the task value function.

### 5.1 TEMPORAL ABSTRACTION AND EXPLORATION

When embedding an action sequence, the DynE objective seeks to preserve information about the outcome of that action sequence (i.e. the change in state), but minimize information about the original action sequence. As shown in Appendix D, this leads to a representation where all action sequences which have similar outcomes embed close together. We propose that this temporally abstract action space, where actions correspond to multi-step outcomes, allows random actions to explore the environment more efficiently.

We empirically validate the exploration benefits of the temporally abstract DynE actions. Figure 3 shows that uniformly sampling a DynE action results in a nearly uniform distribution over the states reachable within $k$ steps. Over the course of an entire episode, selecting DynE actions uniformly at random reaches faraway states more often than random exploration with raw actions. Appendix F shows the qualitative difference between random trajectories in the raw and DynE action spaces, and Appendix C studies the impact of varying $k$ on the performance of a learned policy.

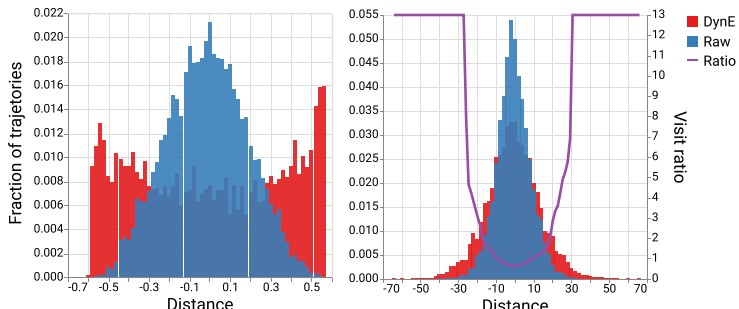

Figure 3: The distribution of state distances reached by uniform random exploration using DynE actions ($k = 4$) or raw actions in Reacher Vertical. **Left:** Randomly selecting a 4-step DynE action reaches a state uniformly sampled from those reachable in 4 environment timesteps. **Right:** Over the length of an episode (100 steps), random exploration with DynE actions reaches faraway states very much more often than exploration with raw actions. The visit ratio shows how frequently DynE exploration reaches a certain distance compared to raw exploration.

### 5.2 STATE REPRESENTATIONS

The DynE objective compresses states while preserving information about the outcome of taking any action in that state. If this compression is successful, states which have similar outcomes will be close together in embedding space. In an MDP, two states which have identical successor states have values which differ by at most the range of the reward function $r_{max} - r_{min}$. While in general states which lead to merely similar successors may have arbitrarily different value, we suggest that in many tasks of interest, similar successors may entail similar value.

We investigate whether the DynE state embedding leads to neighborhoods with similar value in the Reacher Vertical environment. We collect 10K states from a random policy in the environment and perform dimensionality reduction on three representations of those states: the DynE embedding of state images, low-dimensional joint states, and pixels. Figure 4 shows the results of this dimensionality reduction, in which every point is colored by its value under a fully-trained TD3 policy on the low-d states. DynE embeddings have neighborhoods with more similar values than states or pixels.

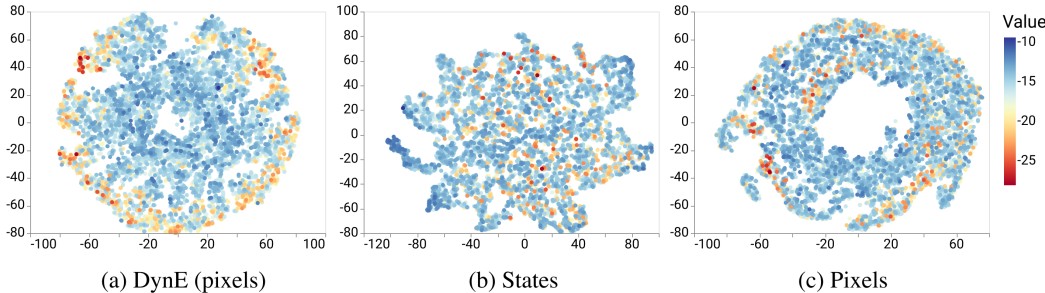

| (a) DynE (pixels) | (b) States | (c) Pixels |

Figure 4: The relationship between state representations and task value. Each plot shows the t-SNE dimensionality reduction of a state representation, where each point is colored by its value under a near-optimal policy. (a) The DynE embedding from pixels places states with similar values close together. (b) The low-dimensional states, which consist of joint angles, relative positions, and velocities, have some neighborhoods of similar value, but also many regions of mixed value. (c) The relationship between the pixel representation and the task value is very complex.

## 6 REINFORCEMENT LEARNING EXPERIMENTS

In this section we assess the effectiveness of the DynE representations for deep RL, individually analyzing the contributions of the action and state representations before combining them. First we evaluate the DynE action space on a set of six tasks with low-dimensional state observations, testing its usefulness across a set of tasks and object interactions. Then, we test the DynE state space on a set of three tasks with pixel observations. Finally, we combine DynE actions with DynE observations, verifying that the two learned representations are complementary.

Appendix B provides a full description of hyperparameters and model architectures, and all of the code for DynE is available on GitHub at `https://github.com/dyne-submission/dynamics-aware-embeddings`.

**Environments** We use six continuous control tasks from two families implemented in the MuJoCo simulator (Todorov et al., 2012) to evaluate our method. Within each family, the task and observation space change but the robot being controlled stays roughly the same, allowing us to test the transferrability of the DynE action space between tasks. The Reacher family consists of three of tasks which involve controlling a 2D, 2DoF arm to interact with various objects. The 7DoF family of tasks from OpenAI Gym (Brockman et al., 2016) is quite difficult, featuring three tasks in which a 3D, 7DoF arm must use different end effectors to push or throw various objects to randomly-generated goal positions. Images and detailed descriptions of both families of tasks are available in Appendix A.

### 6.1 LOW-DIMENSIONAL STATES

For training the DynE action representation we use 100K steps with a uniformly random behavior policy in the simplest environment in each family with no reward or other supervisory signal. As this DynE pretraining is unsupervised and only occurs once for each family of environments, the $x$ axis on these training curves refers only to the samples used to train the policy.[3] We then transfer this action representation to all three environments in the family. When training DynE-TD3 we use all of the default hyperparameters from the TD3 implementation across all environments.

We directly test the impact of switching from raw to DynE actions by comparing TD3 to DynE-TD3. For completeness we compare with two additional state-of-the-art model-free methods: soft actor-

---

[3]On all environments except the simplest (Reacher Vertical) shifting the DynE-TD3 plot by 100K steps does not affect the ordering of the results.

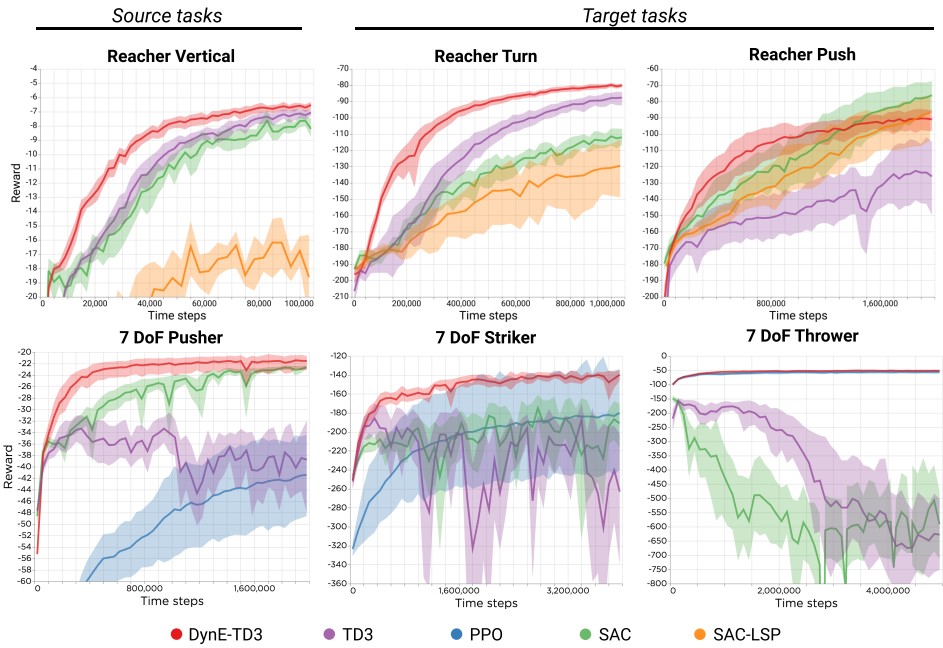

Figure 5: Performance of DynE-TD3 and baselines on two families of environments with low-dimensional observations. Dark lines are mean reward over 8 seeds and shaded areas are bootstrapped 95% confidence intervals. Across all the environments, TD3 learns faster with the DynE action space than with the raw actions. Within each family of environments, the DynE action space was trained only on the simplest task (left).

critic (SAC) (Haarnoja et al., 2018b;c) and proximal policy optimization (PPO) (Schulman et al., 2017). We also compare with soft actor-critic with latent space policies (SAC-LSP) (Haarnoja et al., 2018a), an innovative hierarchical method which transforms a low-level action space into an abstract one by training an invertible low-level policy. In all cases we use the official implementations[4][5][6] and the MuJoCo hyperparameters used by the authors. We also attempted to compare with the hierarchical method by Nachum et al. (2018), but after several emails with the authors and dozens of experiments we were unable to get it to converge on tasks other than those in their paper.

**Results**   Figure 5 shows the results of these experiments. Most significantly, they show that switching from the raw action space (TD3 curve) to the DynE action space results in faster training and allows TD3 to solve the difficult 7DoF suite of tasks. We see that the DynE action space generalizes across several tasks with the same robot, even when interacting with objects unseen during training. It is especially worth noting that the gains from DynE increase as the tasks become harder, maintaining convergence, stability, and low variance in the face of high-dimensional control with difficult exploration. Since SAC-LSP (Haarnoja et al., 2018a) performs similarly but worse than SAC we test it only on the simpler Reacher family of tasks; meanwhile, the PPO curves do not enter the frame on the Reacher family of tasks due to its poor sample efficiency.

## 6.2    PIXELS

Using the Reacher family of environments we evaluate several state representations by their effectiveness for policy learning with TD3.

We evaluate two established methods for learning representations from single images. "DARLA" is the Disentangled Representation Learning Agent proposed by Higgins et al. (2017b) with the denoising autoencoder loss, which is referred to in that work as $\beta$-VAE$_{DAE}$. "VAE" is a standard variational autoencoder (Kingma & Welling, 2013; Rezende et al., 2014), which has previously been

---

[4]TD3: `https://github.com/sfujim/TD3/`

[5]SAC and SAC-LSP: `https://github.com/haarnoja/sac`

[6]PPO: `https://github.com/openai/baselines/tree/master/baselines/ppo2`

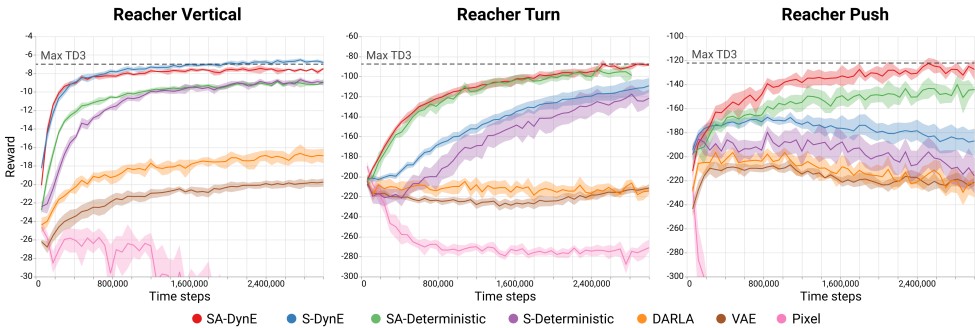

Figure 6: Performance of TD3 trained with various representations. Learned representations for state which incorporate the dynamics make a dramatic difference. SA-DynE converges stably and rapidly and achieves performance from pixels that nearly equals TD3's performance from states. Dark lines are mean reward over 8 seeds and shaded areas are bootstrapped 95% confidence intervals.

found to learn effective representations for control (Van Hoof et al., 2016); it is equivalent to DARLA with the pixel-space loss and $\beta = 1$. Since these representations operate on a single frame at a time, we apply them to the most recent four frames independently and then concatenate the embeddings before feeding them to the policy. These representations have compressed latent spaces, but they encode no knowledge of the environment's dynamics, allowing us to evaluate the importance of incorporating the dynamics into our embeddings.

Next we evaluate representation learning methods whose objectives incorporate the dynamics. "S-DynE," for State DynE, is the DynE state embedding $e_s$, and "SA-DynE" combines the DynE state and action representations. "S-Deterministic" and "SA-Deterministic" are ablations of the corresponding DynE methods which have the same forward-prediction objective but no KL or noise on the latent representations. Comparing the DynE methods to their respective ablations reveals the contribution of explicitly introducing a compression objective to the latent space.

For training all of the learned representations we use a dataset of 100K steps in each environment from a uniformly random policy. In every case we train TD3 with the learned representations using all of the default hyperparameters from the official TD3 implementation.

We compare these representation learning methods with TD3 trained from pixels. As there are no experiments on pixels in the TD3 paper, we performed extensive search over network architectures and hyperparameters. We included in our search the configurations used in the pixel experiments of DDPG (Lillicrap et al., 2015) as well as those used in successful discrete-action RL works from pixels (Schulman et al., 2017; Kostrikov, 2018; Espeholt et al., 2018).

**Results** Figure 6 shows the results of these experiments. We find that the single-image methods are unable to solve any of the three tasks from pixels; TD3 from pixels diverges in all cases, while VAE and DARLA learn gradually at best. If simply reducing the dimension of the states were sufficient to enable effective policy training, we would expect good performance from these methods. S-DynE and S-Deterministic, which incorporate the dynamics into their representation learning objectives, perform far better. The minimality imposed by the DynE objective allows S-DynE and SA-Dyne to outperform their deterministic ablations. SA-DynE learns rapidly and reliably, finding behaviors which qualitatively solve all three tasks. The improvement of SA-DynE over S-DynE shows that the state and action representations are complementary.

## 7 DISCUSSION

In this work we proposed a method, Dynamics-aware Embedding (DynE), that jointly learns embedded representations of states and actions for reinforcement learning. Our experiments reveal that DynE action embeddings lead to more efficient exploration, resulting in more sample efficient learning on complex tasks, while DynE state embeddings allow unmodified model-free RL algorithms to scale to pixel observations. When combined, the DynE state and action embeddings result in stable, sample-efficient learning of high-quality policies from pixels.

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

## APPENDIX A  ENVIRONMENT DESCRIPTION

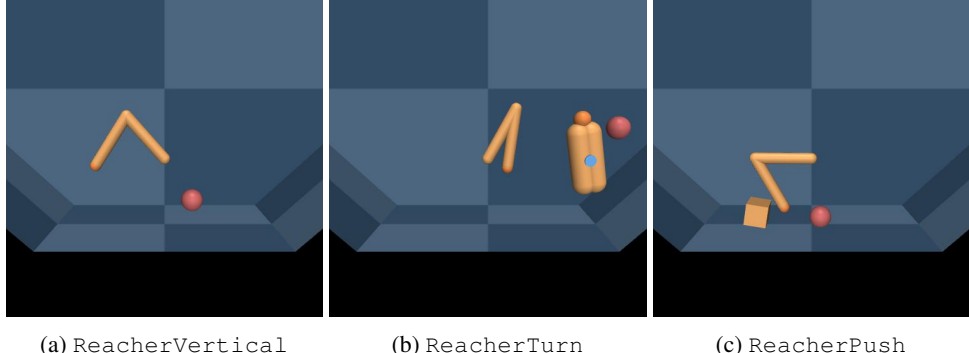

(a) `ReacherVertical`          (b) `ReacherTurn`          (c) `ReacherPush`

Figure 7: The Reacher family of environments. `ReacherVertical` requires the agent to move the tip of the arm to the red dot. `ReacherTurn` requires the agent to turn a rotating spinner (dark red) so that the tip of the spinner (gray) is close to the target point (red). `ReacherPush` requires the agent to push the brown box onto the red target point. The initial state of the simulator and the target point are randomized for each episode. In each environment the rewards are dense and there is a penalty on the norm of the actions. The robot's kinematics are the same in each environment but the state spaces are different.

The first task family, pictured in Figure 7, is the "Reacher family", based on the `Reacher-v2` MuJoCo (Todorov et al., 2012) task from OpenAI Gym (Brockman et al., 2016). These tasks form a simple new benchmark for multitask robot learning. The first task, which we use as the "source" task for training the DynE space, is `ReacherVertical`, a standard reach to a location task. The other two tasks are inspired by the DeepMind Control Suite's `Finger Turn` and `Stacker` environments, respectively (Tassa et al., 2018). In `ReacherTurn`, the same 2-link Reacher robot must turn a spinner to the specified random location. In `ReacherPush`, the Reacher must push a block to the correct random location.

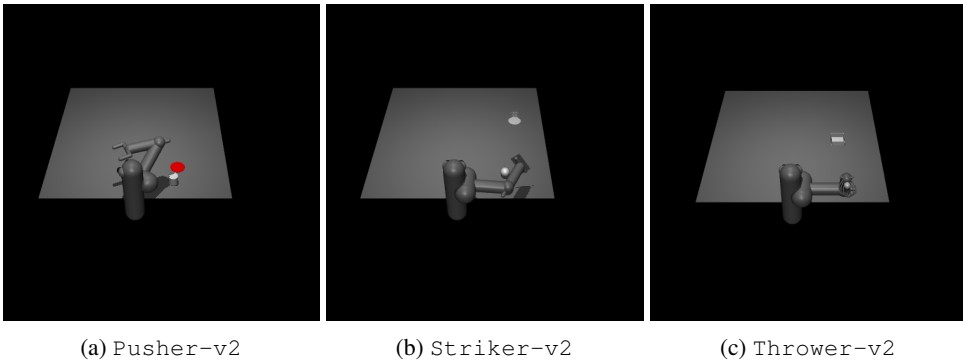

(a) `Pusher-v2`          (b) `Striker-v2`          (c) `Thrower-v2`

Figure 8: The 7DoF family of environments. `Pusher-v2` requires the agent to use a C-shaped end effector to push a puck across the table onto a red circle. `Striker-v2` requires the agent to use a flat end effector to hit a ball so that it rolls across the table and reaches the goal. `Thrower-v2` requires the agent to throw a ball to a target using a small scoop. As with the Reacher family, the dynamics of the robot are the same within the 7DoF family of tasks. However, the morphology of the robot, as well as the object it interacts with, is different.

The second task family is the "7DoF family", which comprises `Pusher-v2`, `Striker-v2`, and `Thrower-v2` from OpenAI Gym (Brockman et al., 2016). We use `Pusher-v2` as the source task. These tasks use similar (though not identical) robot models, making them a feasible family of tasks for transfer. They are shown in Figure 8.

## A.1 Pixels

We use full-color images rendered at 256x256 and resized to 64x64 pixels. In order to allow the agents to perceive motion, we stack the current frame with the three most recent frames, resulting in an observation of dimension 12x64x64.

## Appendix B  Hyperparameters and DynE training

For DynE-TD3 we use all of the default hyperparameters from the TD3 code[7] across all tasks. For all experiments we choose the dimension of the DynE action space to be equal to the dimension of a single action in the environment. We set the number of actions in the DynE space to be $k = 4$ for all experiments except `Thrower-v2`, for which we use $k = 8$. We use the Adam optimizer (Kingma & Ba, 2014) with learning rate $10^{-4}$. All our experiments used recent-model NVidia GPUs.

**Training on states**   When computing log-likelihoods we divide by the number of dimensions in the state in an attempt to make the correct settings of $\gamma$ invariant to the observation dimension; the same result could be achieved by multiplying the values of $\gamma$ that we report by the state dimension and changing the learning rate. With that scaling we set we set our hyperparameters $\gamma = \lambda = 10^{-2}$ across all environments. We concatenate all the joint angles and velocities to use as the states during representation learning. We preprocess the $s, s'$ pairs by first taking the difference $\Delta s = s' - s$ and then whitening so that $\Delta s$ has zero mean and unit variance in each dimension. This preprocessing encourages the encoder to represent both position and velocity in the latent space; the scales of these two components are quite different.

We use fully-connected networks for the action encoder $e_a$ and the conditional state predictor $f$. Each function has two hidden layers of 400 units. Training this model should take 5-10 minutes on GPU.

**Training on pixels**   We train a DynE model for each environment, taking in a stack of frames and a sequence of $k = 4$ actions and predicting future states. To speed training we predict only the two latest frames of the future state (i.e. the picture of the world at time $t + k$ and $t + k - 1$) instead of all four. When doing RL we take the state encoder $e_s$ from this model and use it to preprocess all states from the environment.

We set the dimension of the state embedding $z_s$ to 100. We did not try other options, and given the sensitivity of RL to state dimension a smaller setting would very likely yield faster learning. We set $\beta = \gamma = 1$, at which setting DynE is optimizing a variational lower bound on $p(s_{t+k}|s_t, \boldsymbol{a}^k)$. We recommend ensuring that the predictions (not generations) from the model are correctly rendering all the task-relevant objects; if $\beta$ and $\gamma$ are too high, the model may incur lower loss by ignoring details in the image. We use cyclic KL annealing (Liu et al., 2019) to improve convergence over a wide range of settings.

We use the DCGAN architecture (Radford et al., 2015) for the image encoder $e_s$ and the predictor $f$. The action encoder $e_a$ is fully connected with two hidden layers of 400 units. Training this model takes 1-2 hours on GPU.

---

[7]`https://github.com/sfujim/TD3`

## APPENDIX C    VARYING LEVELS OF TEMPORAL ABSTRACTION

We study the impact of varying $k$, the level of temporal abstraction in the DynE action space. We find that increasing $k$ improves performance and learning speed up to a point; beyond this point, performance degrades. The optimal setting of $k$ will depend on the environment dynamics. We expect that environments with very slow dynamics will benefit from a greater degree of temporal abstraction.

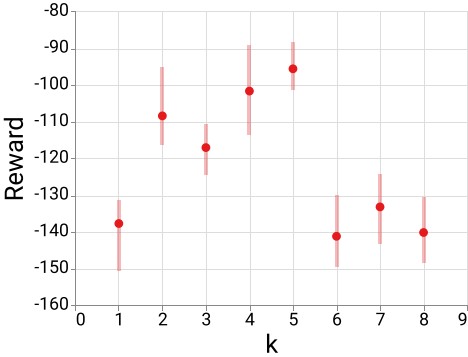

Figure 9: DynE-TD3 results on Reacher Push with varying $k$. We find that increased temporal abstraction improves performance up to a point, beyond which the action space is no longer able to represent the optimal policy and performace degrades. Solid points are the mean reward obtained after training for 1M environment steps. Shaded bars represent the min and max performance over 4 seeds.

## APPENDIX D    VISUALIZING THE DYNE ACTION SPACE

To better understand the structure in the DynE action embedding space, we visualize the relationship between the outcome of a sequence of actions and the DynE embedding of those actions. When embedding an action sequence, the DynE objective seeks to preserve information about the outcome of that action sequence (i.e. the change in state), but minimize information about the original action sequence. Therefore we should see that all action sequences which have similar outcomes embed close together, regardless of the actions along the way. Figure 10 investigates this in a simple Point environment with an easy-to-visualize 2D $(x, y)$ state. For this simple problem, we see that all pairs of action sequences $\boldsymbol{a}_1^k$ and $\boldsymbol{a}_2^k$ with similar outcomes are close together in the embedding space. The correspondence between the two spaces appears to remain strong for high-dimensional and nonlinear environments, but is much harder to render in two dimensions.

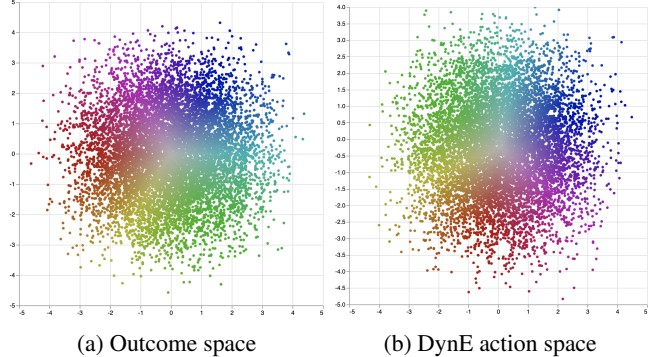

(a) Outcome space          (b) DynE action space

Figure 10: The mapping between the outcomes and embeddings of action sequences. We sample 10K random sequences of four actions and evaluate their outcomes in the environment dynamics, measured by $(\Delta x, \Delta y) = s_{t+4} - s_t$. **(a)** We plot the outcome $(\Delta x, \Delta y)$ of each action sequence and color each point according to its location in the plot. **(b)** We use DynE to embed each action sequence into two dimensions; each point in this plot corresponds to a point in (a) and takes its color from that corresponding point. The similarity of the two plots and the smooth color gradient in (b) indicate that DynE is embedding action sequences according to their outcomes.

## APPENDIX E   EXTENDED RESULTS

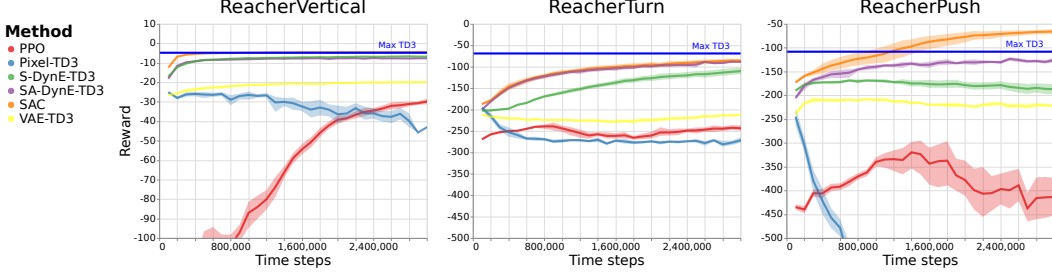

Figure 11: These plots allow for direct comparison between the methods from pixels (Pixel-TD3, VAE-TD3, S-DynE-TD3, and SA-DynE-TD3) and our baselines from low-dimensional states (PPO and SAC). The DynE methods from pixels perform competitively with some baselines from states.

## APPENDIX F    EXPLORATION WITH RAW AND DYNE ACTION SPACES

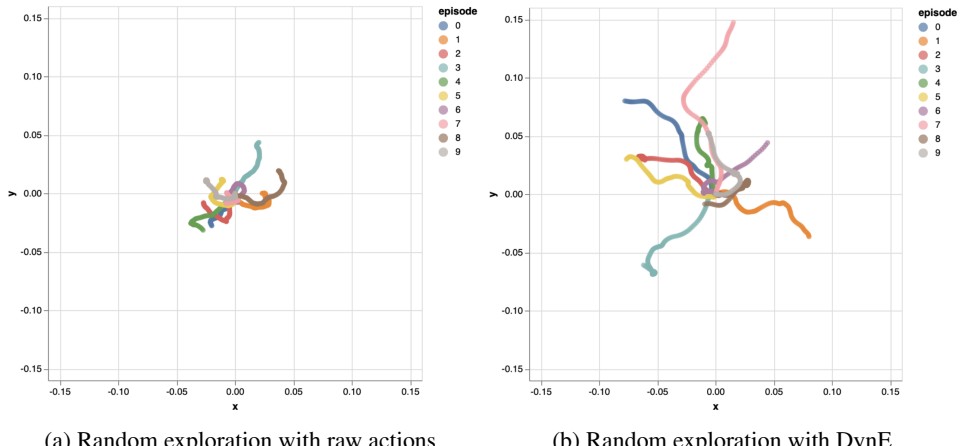

(a) Random exploration with raw actions          (b) Random exploration with DynE

Figure 12: These figures illustrate the way the DynE action space enables more efficient exploration. Each figure is generated by running a uniform random policy for ten episodes on a `PointMass` environment. Since the environment has only two position dimensions, we can plot the actual 2D position of the mass over the course of each episode. **Left:** A policy which selects actions at each environment timestep uniformly at random explores a very small region of the state space. **Right:** A policy which randomly selects DynE actions once every $k$ timesteps explores much more widely.

