# OpenReview forum: "Dynamics-Aware Embeddings"
_ICLR.cc/2020/Conference — Accept (Poster)_

### Official Review · AnonReviewer1 · 2019-10-23
**Official Blind Review #1**

**Rating:** 8

**Review:**

This paper presents DynE, a self-supervised approach for learning dynamics-aware state and action representations. DynE learns an encoding of individual states and action sequences, assigning nearby embeddings to those that have similar outcomes. This is achieved via a reconstruction objective that predicts the outcome of a sequence of "k" actions from a given state, along with losses that encourage compression in the learned latent representations. Additionally, a learned decoder allows for reconstruction of minimum-norm action sequences from the high-level latent action embedding. Combining DynE state and action embeddings with an actor-critic agent operating directly in the learned high-level action space leads to significant speedups in both from-scratch and transfer learning (on 2D and 3D OpenAI Gym tasks), leading to better data efficiency compared to model-free baselines. Additionally, the learned action and state embeddings lend themselves to better exploration and consistent value prediction, respectively.

The paper is very well written and the approach looks quite promising. A few comments:
1. The approach is well validated but additional ablation results can help quantify the effect of different components. For example, it would be useful to see the effect of varying "k", the number of actions to be encoded for generating the action embedding.
2. A related paper that learns state representations that are physically consistent and dynamics-aware is this work:
Jonschkowski, Rico, et al. "Pves: Position-velocity encoders for unsupervised learning of structured state representations." arXiv preprint arXiv:1705.09805 (2017).
Here the state representation is learned to implicitly encode physical consistency via self-supervised losses that mimic constraints such as controlability, inertia, conservation of mass etc. Combining such additional self-supervised losses can help structure the state embedding learning further, albeit at the cost of introducing additional hyperparameters during optimization.
3. It would be useful to know what the actions are (and their dimensions) for the tasks considered in the paper.
4. The paper would benefit from a short discussion on the limitations of the proposed approach and potential to scale to more complicated tasks.
5. Fig. 5, bottom right: It is not clear why PPO (blue) performs significantly better on this task compared to the other 7DoF tasks considering that the thrower should be more complex than the pusher and striker. PPO also seems to match the data efficiency of DynE-TD3. Is this correct?

Overall, I find the approach quite interesting and promising. I would suggest an accept.

Typos:
1. Intro, 2nd para, 2nd line, many samples to learn than a better one
2. Fig. 1, the pixel representation is very unintuitive

**Experience Assessment:**

I have read many papers in this area.

**Review Assessment: Checking Correctness Of Derivations And Theory:**

I assessed the sensibility of the derivations and theory.

**Review Assessment: Checking Correctness Of Experiments:**

I assessed the sensibility of the experiments.

**Review Assessment: Thoroughness In Paper Reading:**

I made a quick assessment of this paper.

---

> ### Author Response · Authors · 2019-11-14
> **Response to Reviewer #1**
>
> Thank you for your review!
>
> We agree that a study of performance with varying $k$ is useful and we have added one as a new Appendix C. We find that there is an optimal setting of $k$ which is large enough to enable efficient exploration while still representing the optimal policy with high fidelity.
>
> The "Pves" method of Jonschkowski et al. is an interesting approach and we have added it to our related work. Thanks for the reference!
>
> The actions in each environment set the torque of the actuators. For each environment there is one action dimension for each joint, i.e. 2D actions for the Reacher family and 7D actions for the 7DoF family. The action scales are bounded within [-1, 1]. In our approach we set the embedded action dimension to be the same as the raw action dimension. We will make this more clear in Appendix A.
>
> At present the greatest limitation of our approach as described is the pretraining on a fixed dataset. We chose to use a fixed dataset to disentangle the tasks of representation learning and policy learning, and to simplify comparisons with the other representation learning methods in section 6.2. However, DynE is compatible with online learning and one can use an exploration strategy from the literature to collect data or even modify our approach as follows:
> 1. Add each transition observed to the representation learning dataset and periodically retrain $e_s$, $e_a$, and $d_a$ according to sections 2.2 and 3.1.
> 2. When updating the policy and Q function, recompute the embedded states $e_s(s)$ using the updated state encoder $e_s$.
> 3. When updating the policy and Q function, the encoded actions $z_t$ which were emitted by the policy may now map to a different sequence of actions $d_a(z_t) = a_t, \dots, a_{t+k-1}$ than when that transition was added to the replay, making an update using $z_t$ incorrect. Instead we must re-encode the actions in the replay at policy update time: $z_t = e_a(a_t, \dots, a_{t+k-1})$.
>
> We also found PPO's performance on Thrower surprising. Thrower appears to be a fairly different task than Pusher and Striker; DynE-TD3 and PPO solve Thrower quite quickly, while TD3 and SAC fail entirely. We also note that the scale of the Thrower plot is distorted by the divergence of TD3 and SAC, and zooming in reveals that DynE-TD3 converges to a better solution than PPO: https://i.imgur.com/527l9hZ.png

---

### Official Review · AnonReviewer3 · 2019-10-23
**Official Blind Review #3**

**Rating:** 6

**Review:**

The author propose a representation learning method based on predictive information. They compress a start state and an action sequence to predict the following state. Since the latent space is factorized between state and action sequence, it can be used as an abstract action space to accelerate model-free algorithms.

Strengths:
- While the state representation is a simple successor representation, the action abstraction is a simple method that seems novel.
- The multi-step return is a nice way of handling variable horizons in the context of temporally abstract actions.
- It is nice to see that the representation learning method can accelerate learning not just from pixels but also when learning from low-dimensional inputs.
- The method description and overall writing is very clear.

Weaknesses:
- Doesn't the multi-step return render the update on-policy, since the reward sequence is tied to the data collecting policy? If so, it might be worth to apply off-policy corrections from the literature. If not, this should be explained in Section 3.2.
- A comparison across more domains would be desirable. While there are 6 visual tasks, they share only two environments. The paper could be strengthened by comparison on standard benchmarks such as Gym or DMControl. I'm willing to raise my score when these or comparable results are added.
- I could not find a clear description of how the hyper parameters of baseline methods were selected, so it is unclear how much of the benefit comes from tuning.

Comments:
- Equation numbers are missing on page 4.
- An assumption of the work is that the pixel observations are Markovian. Maybe I missed this in the paper, but was there any frame stacking that would make this hold at least approximately?

**Experience Assessment:**

I have published in this field for several years.

**Review Assessment: Checking Correctness Of Derivations And Theory:**

I carefully checked the derivations and theory.

**Review Assessment: Checking Correctness Of Experiments:**

I assessed the sensibility of the experiments.

**Review Assessment: Thoroughness In Paper Reading:**

I read the paper thoroughly.

---

> ### Author Response · Authors · 2019-11-14
> **Response to Reviewer #3**
>
> Thanks for the review!
>
> Because of the temporally abstract actions the multi-step update is off-policy. Conditioning the critic on the latent action $z_t$ and the current step $i$ of that action is equivalent to conditioning on the remaining raw actions $a_t, \dots, a_{t+k-i}$ of that sequence. Given this information the probability of those next $k-i$ actions is 1 and no off-policy correction is needed; in effect, it remains a single action. Thank you for pointing out that this was not made explicit in the paper. We have added this to section 3.2.
>
> We agree that a comparison on more domains would be useful and we will add one to the final version of this paper, though this experiment may not be completed by the end of this discussion period due to computational cost (each of the three plots in Figure 6 corresponds to > 1000 GPU-hours). We also note that Pusher, Striker, and Thrower are standard benchmarks from OpenAI Gym.
>
> All of the model-free baselines have previously been tested on MuJoCo environments with hyperparameters selected by their authors. We use the MuJoCo hyperparameters from the papers or implementations of each method we compare to. Across all of the representation learning results which use TD3, including DynE-TD3 in Figure 5 and all methods in Figure 6, we use the hyperparameters from the original TD3 paper without modification.
>
> Re: Markov observations, we stack four frames to ensure that the Markov property holds (Appendix A.1).

---

> > ### Comment · AnonReviewer3 · 2019-11-15
> > **Off-policy correction**
> >
> > Thank you for the clarifications. My understanding is that the abstract action space is learned over time and thus keeps changing. Is this correct? In that case, it seems like the low-level action sequence corresponding to a high-level action would change over time. This could be corrected for e.g. using the approach of HIRO (Nachum et al. 2018).

---

> > > ### Author Response · Authors · 2019-11-15
> > > **Re: off-policy correction**
> > >
> > > In this work the abstract action space is learned ahead of time, though this need not be true in general. When learning the action space online, you are exactly right that the shifting map between abstract and raw actions needs to be corrected. HIRO is a good option for this. Since we have an encoder $e_a$ which maps from raw actions to abstract actions, we could also relabel the upcoming set of $k$ actions as $\tilde{z}_a = e_z(a_{t:t+k-1})$. This should improve on the expense and bias of HIRO's sampling-based relabeling (Appendices A and C.3 of Nachum et al. 2018).

---

> > > > ### Comment · AnonReviewer3 · 2019-11-15
> > > > **Thanks**
> > > >
> > > > Thank you for the clarification.

---

### Official Review · AnonReviewer4 · 2019-10-26
**Official Blind Review #4**

**Rating:** 8

**Review:**

This paper presents an approach to learning state and action representations through self-supervision, such that these representation can be used for downstream reinforcement learning. In particular, the proposed approach learns a  time-dilated dynamics model on data collected via self-supervision, where given s_t, and actions (a_t, ..., a_{t+K}) predicts s_{t+K}. The input state and action trajectory and each encoded into latent distributions, which are then used to reconstruct the future state. Then, they demonstrate that using TD3 with the latent action space outperforms existing model-free methods and existing state representation techniques.

Overall the paper is well motivated and clearly written. The key contribution seems to be in learning the latent distribution over multi-step action trajectories, which seems to be important for performance. Lastly the experiments and ablations are thorough and well explained.

My main comments have to do with (1) the fairness of the comparison to existing model-free RL, (2) an analysis of the temporal abstraction for learning the action distribution.

(1): The proposed method first pre trains the latent dynamics model on 100K steps of random data, then trains the proposed TD3 using this action distribution (and the modified critic to support 1-step Q values). While this does outperform the model-free RL methods trained from scratch, it is also using 100K steps worth of experience that the others don't have access to, which makes it not quite a fair comparison. If you pretrain the critic of TD3 or SAC with the 100K samples, do you still observe the same performance gains?

(2): From the ablation study and comparison to other state representation learning techniques in Figure 6, it seems like the most important aspect of the proposed method is using the latent action distribution. This makes sense as it captures longer action sequences, and thus likely is the reason for better exploration and performance. As a result the exact choice of K seems very important. In the Appendix it states that for the Thrower task K=8, and elsewhere K=4. Do the authors have a sense for how performance changes with choice of K? I think a plot which compares performance over different choices of K would be very valuable.

Some smaller comments:
- The comparison to other model-free RL methods is done only on low dimensional states, while the ablations are done on pixels. Is this because the model-free comparisons did not work at all on pixels?
- Is it possible to perform model predictive control with the learned model, and how does it compare to existing latent model based RL methods (Hafner et al.)
- One more recent work that may be worth comparing to is SLAC (Lee et al.) which also learns a stochastic latent dynamics model, and learns a policy in the latent space of the model. The latent space is of states however, and not actions.

______________________

Alex X. Lee, Anusha Nagabandi, Pieter Abbeel, Sergey Levine. Stochastic Latent Actor-Critic: Deep Reinforcement Learning with a Latent Variable Model

Danijar Hafner, Timothy Lillicrap, Ian Fischer, Ruben Villegas, David Ha, Honglak Lee, James Davidson. Learning Latent Dynamics for Planning from Pixels


**Experience Assessment:**

I have published one or two papers in this area.

**Review Assessment: Checking Correctness Of Derivations And Theory:**

I assessed the sensibility of the derivations and theory.

**Review Assessment: Checking Correctness Of Experiments:**

I carefully checked the experiments.

**Review Assessment: Thoroughness In Paper Reading:**

I read the paper at least twice and used my best judgement in assessing the paper.

---

> ### Author Response · Authors · 2019-11-14
> **Response to Reviewer #4**
>
> We appreciate the comments!
>
> Regarding the samples used for representation learning, the simplest comparison is to offset the DynE curves by 100K steps, including the random transitions in their sample cost. In all environments but the simplest (Reacher Vertical), the policies trained with DynE still learn faster than the baselines (see footnote on page 6). Note also that in Figure 5, DynE only pretrains on data from the leftmost task in each row, demonstrating that transfer further improves its sample efficiency.
>
> The latent action representation does provide significant gains on top of the learned state representation. However, we also find substantial gains from the DynE state representation, as shown by the gap between S-DynE and DARLA. We observe that the improvement from DARLA to S-DynE is similar in scale to the improvement from S-DynE to SA-DynE.
>
> We agree that a study of performance with varying $k$ is useful and we have added one as a new Appendix C. We find that there is an optimal setting of $k$ which is large enough to enable efficient exploration while still representing the optimal policy with high fidelity.
>
> We chose to compare to TD3 on pixels because it allowed for the most direct comparison to our results and none of the model-free methods work well from pixels anyway. As the pixel experiments are quite computationally intensive to run we found it more informative to compare against other representation learning algorithms.
>
> In principle one could perform MPC with this learned model. However, we would not expect it to perform well as our objective is designed to induce useful representations and not to make accurate long-term predictions. In particular, successful model-based RL methods like Hafner et al. (2019) and Chua et al. (2018) directly optimize their models with multi-step prediction objectives, with Hafner et al. making multi-step predictions without decoding back to observations.
>
> We do think combining learned temporally abstract action representations with MPC is an interesting future direction as it would allow more efficient rollouts and planning.
>
>
> Hafner, Danijar, et al. "Learning Latent Dynamics for Planning from Pixels." International Conference on Machine Learning. 2019.
> Chua, Kurtland, et al. "Deep reinforcement learning in a handful of trials using probabilistic dynamics models." Advances in Neural Information Processing Systems. 2018.

---

> ### Comment · AnonReviewer4 · 2019-11-15
> **Response to rebuttal**
>
> Thank you for the comments! You have addressed both my main concerns, and I think the added Appendix C is quite interesting. I wonder if learning the right value of k is a direction for future work.
>
> I am increasing my score to "Accept"

---

### Official Review · AnonReviewer2 · 2019-11-03
**Official Blind Review #2**

**Rating:** 3

**Review:**

Summary: The paper proposes training dynamics-aware embeddings of the state and k-action sequences to aid the sample efficiency of reinforcement learning algorithms. The authors propose learning a low-dimensional representation of the state space, $z_s$ as a well as a temporally extended action embedding, $z_a$. The latter will be used in conjunction with a higher level policy that plans in this abstract action-space, $z_a$. By using these two embeddings, the authors test the proposed system on a set of Mujoco tasks and show improved results.

Positives:
1) Fairly simple objective, in line with previous work on unsupervised learning methods to representation learning in RL (like DARLA, variational intrinsic control, etc).
2) The temporally extended nature of the action embedding makes is particularly attractive for HRL systems as a continuous space of options (via $z_a$).



Questions and Points of improvements:
1) Main concern: The need to pre-train the embedding before the RL task, I strongly believe limits the applicability of the proposed algorithm. The embeddings are trained under a uniformly random policy, which in many cases in RL is not informative enough to reach, with decent probability, many of the states of interest. Thus the embedding will reflect only a small subset of the state/action-space. Thus it will be highly depend on the tasks under consideration if this is enough variety for generalisation across the stationary distribution of more informed RL policies. Implicitly, the authors are making a continuity assumptions over the state and action space.
(To be more precise: A particular failure case of the action embedding would be if say one of the action (down) has no effect in the part of the space where the uniform policy has explored. Now this becomes an important action in a level down the line where the agents needs to go down a tunnel -- example from Atari's Pitfall. In this case, under the embedding training, since the down has had no effect in training, this action will not be represented at all. This would means the RL algorithm could not ever learn to use it).
The co-evolution of the representation and the RL policy, I think it's paramount especially when dealing with exploration.

2) Q: Section 3.2: "we extend... to work with temporally extended actions while maintaining off-policy updates ..". Can the authors expand on how this is done? Both updates in this section seem to be on policy ($\mu$).

3) Q: Section 3.2: "Q can only be trained on $N/k$ observations. This has a substantial impact on sample efficiency". Note that this is actually an explicit trade-off between reduced number of samples we see ($N/k$) and the increased horizon in propagating information, due to the effective k step update. This trade-off need not be optimal for $k=1$.

4) Notes of experiments:
a) It is hard to assess the difficulty of the exploration problems investigated. This relates to point 1) and the implicit assumptions highlighted there.
b) It would have been nice to have a study on $k$ and it's impact on the sample complexity. The larger the $k$ the harder the representation learning problem becomes; and possibly the larger the number of samples needed to learn in this combinatoric space. How does this trade-off with the benefits one could potentially get in the RL phase?
c) For the comparison algorithms: where any of these using a temporal extended update rule?  Or are all of them 1-step TD like algorithms? It would good to separate the effect of the multiple-step update in Sec. 3.3 and the exploration in this abstract action space.






**Experience Assessment:**

I have read many papers in this area.

**Review Assessment: Checking Correctness Of Derivations And Theory:**

I assessed the sensibility of the derivations and theory.

**Review Assessment: Checking Correctness Of Experiments:**

I assessed the sensibility of the experiments.

**Review Assessment: Thoroughness In Paper Reading:**

I read the paper at least twice and used my best judgement in assessing the paper.

---

> ### Author Response · Authors · 2019-11-14
> **Response to Reviewer #2**
>
> Thank you for your comments!
>
> Regarding the pre-training of embeddings, in this work the main focus is on the objectives used for representation learning. Separating exploration and representation learning allows us to directly compare the various representation learning techniques in section 6.2. Once we have such an objective, exploration and representation learning can be combined online. Our method is compatible with such online representation learning. One specific implementation would involve the following steps:
> 1. Add each transition observed to the representation learning dataset and periodically retrain $e_s$, $e_a$, and $d_a$ according to sections 2.2 and 3.1.
> 2. When updating the policy and Q function, recompute the embedded states $e_s(s)$ using the updated state encoder $e_s$.
> 3. When updating the policy and Q function, the encoded actions $z_t$ which were emitted by the policy may now map to a different sequence of actions $d_a(z_t) = a_t, \dots, a_{t+k-1}$ than when that transition was added to the replay, making an update using $z_t$ incorrect. Instead we may re-encode the actions in the replay at policy update time: $z_t = e_a(a_t, \dots, a_{t+k-1})$.
> With these modifications it is possible to learn the representations and policy at the same time.
>
> The updates in Section 3.2 are off-policy because they depend on the current policy $\mu$, but crucially not on the behavior policy $\pi$ which collected the data. This is the same for all algorithms in the DPG family. See Silver et al. (2014) for details, especially section 4.2.
>
> As you point out, updating on only $N/k$ observations in the abstract MDP might outperform learning in the original MDP despite having fewer samples. However, as we show in section 3.2, we can update on all $N$ samples while still using the embedded MDP by augmenting Q with an abstract step input $i$.
>
> We agree that a comparison of performance with varying $k$ is useful and we have added one as a new Appendix C. We find that increasing $k$ helps up to a certain point, beyond which performance falls off.
>
> Multi-step baseline updates: PPO uses generalized advantage estimation (Schulman et al. 2015), a multi-step return estimator similar to TD($\lambda$). TD3 and SAC use one-step returns. In the off-policy setting, unweighted multi-step returns are not guaranteed to converge (Harutyunyan et al. 2016), and techniques such as importance weighting are not available with deterministic policies like TD3 (as the density is a delta function).
>
>
> Silver, David, et al. "Deterministic Policy Gradient Algorithms." International Conference on Machine Learning. 2014.
> Schulman, John, et al. "High-dimensional continuous control using generalized advantage estimation." arXiv preprint arXiv:1506.02438 (2015).\
> Harutyunyan, Anna, et al. "Q($\lambda$) with Off-Policy Corrections." International Conference on Algorithmic Learning Theory. Springer, Cham, 2016.

---

### Decision · Program_Chairs · 2019-12-19

**Decision:**

Accept (Poster)

**Comment:**

This paper studies how self-supervised objectives can improve representations for efficient RL. The reviewers are generally in agreement that the method is interesting, the paper is well-written, and the results are convincing. The paper should be accepted.